# The mental health impact of physical inactivity: A study on UAE adolescents

Faten Mahmoud Diab[1], Shatha AL-Sharbatti[2], Abdalla Tamer Eltanbadawy[3], Rami Aws Alfahad[4], Ghada Elsayed Elgarawany [1,5]*

1 Biomedical Science Department, College of Medicine, Gulf Medical University, Ajman, United Arab Emirates, 2 Community Medicine Department, College of Medicine, Gulf Medical University, Ajman, United Arab Emirates, 3 Sharjah British International School, Sharjah, United Arab Emirates, 4 Jumeirah English Speaking School Dubai, Dubai, United Arab Emirates, 5 Department of Medical Physiology, Faculty of Medicine, Menoufia University, Shebeen Elkom, Egypt

* dr.ghada@gmu.ac.ae, ghadagrawany@yahoo.com

## Abstract

Mental and physical health are essential for well-being, particularly during adolescence. The study aimed to assess the prevalence, factors, and relationship between physical inactivity and depression, and to determine the predictors of depression among adolescents in the UAE. A cross-sectional study was conducted among students in grades 9–12 from selected schools in the UAE. The study included adolescents of all genders and all nationalities. Two standardized questionnaires were employed: the "Physical Activity Questionnaire for Adolescents (PAQ-A)" to evaluate physical activity levels and the "Centre for Epidemiologic Studies Depression Scale (CES-D)" to assess symptoms of depression among the participants. Correlation test, independent t-test, chi-square test, bivariate and multivariate logistic regression analyses were used, with significance set at $p < 0.05$. The study included 365 participants, with the majority being over 14 years old (68.7%), female (60%), and in grade 10 (32.2%). The prevalence of depression and physical inactivity was 51.2% and 68.8%, respectively, and a significant negative correlation was found between depression and physical activity scores. Significant positive correlations are found between depression and grade level, and chronic health problems, and a negative correlation with crowding index. Significant negative correlations are found between physical activity, age, grade level, and chronic health problems. Predictors of depression are physical inactivity, female gender, overweight and obesity, and having a chronic disease. The study reveals a high prevalence of depression and physical inactivity among adolescents in the UAE. Consequently, physical activity may serve as a key protector against depression, and it potentially influences mental health through many physiological, neurological, and behavioural pathways. This underlines the critical need for interventions aimed at promoting physical activity.

**Data availability statement:** All relevant data are within the manuscript and its Supporting information files.

**Funding:** The author(s) received no specific funding for this work.

**Competing interests:** The authors have declared that no competing interests exist.

**Abbreviations:** PAQ-A, Physical Activity Questionnaire for Adolescents; CES-D, Centre for Epidemiologic Studies Depression Scale

## Introduction

Adolescence represents a critical period of development during which personal lifestyle choices and behaviour patterns are established, including the choice to be physically active or inactive [1]. The World Health Organization has suggested that the daily recommendation for children and adolescents (ages 5–17 years) should include 60 minutes of moderate-to-vigorous intensity physical activity (MVPA). Insufficient physical activity in adolescents is a severe problem worldwide with 81% of adolescents, between the ages of 11 and 17 years old, not fulfilling these daily recommendations. In the United States, the numbers are similar to global rates. Interestingly, these low physical activity levels appear with age from childhood to adolescence, with sharp decreases typically seen between 9 and 15 years of age [2].

In UAE, the overall physical activity levels remain low, and sedentary behaviours remain high among UAE children. In 2016, only 16% of UAE children achieved the recommended amount of MVPA (i.e., ≥ 60 min/day), reflecting that the majority of the UAE children are not achieving the daily recommendations for physical activity [3].

Adolescence is a critical period in human development, marked by significant physical, emotional, and psychological changes. A lot of physiological and social factors can increase the prevalence of mental health problems during this period [4]. According to the WHO, globally, one in seven adolescents experiences a mental disorder, yet these remain largely unrecognized and untreated [2]. Wilson and Dumornay found that rates of depression increased from 2009 to 2019, among both girls and boys, but the percentage change was larger for girls (12.0%) than for boys (3.7%) [5].

A nationwide cross-sectional study that included 9856 adolescent boys and girls who were selected from various divisions in Bangladesh showed that the prevalence of no or minimal, mild, moderate, moderately severe and severe depression was 75.5%, 17.9%, 5,4%, 1.1% and 0.1%, respectively [6].

A cross-sectional study was conducted among 542 randomly selected school-going adolescents, from eight government and private schools in Chandigarh, India, by multistage sampling technique. The authors found that 40% of adolescents had depressive disorders, 7.6 percent major depressive disorders, and 32.5 percent other depressive disorders [7].

A cross-sectional study included 312 higher secondary school students randomly sampled from four schools in Pokhara Metropolitan, Nepal. The authors found that 44.2% of students were depressed, 25.3% of the students were noted to have mild depression and 18.9% of the students expressed major depression [8].

Evidence suggests physical activity is a protective factor against mental health problems such as depression [9]. Several researchers reported that adolescents who engaged in physical activity experienced fewer mental health problems; they showed a negative association between physical activity with anxiety and depression among children and youth [10–12].

Cohort study done by Bell et al found no strong evidence that physical activity is associated with better mental wellbeing or decreased mental health disorder in

adolescents. However, physical activity has the potential to reduce symptoms of anxiety and depression in adolescents, as they found a protective association between physical activity and the emotional problems subscale of Strength Difficulty Questionnaire [13]. Ellis et al. concluded that engagement of youth in physical activity has a protective factor during COVID-19, This study was conducted on Canadian youth ages 14–18. The results showed COVID-19 stress was related to more loneliness and more depression, especially for youth who spend more time on social media. Spending time with family, connecting to friends, and physical activity were related to lower loneliness related to COVID-19 stress [14].

A previous study from UAE among 600 adolescents detected depressive symptoms among 17.2% of participants [15]. Results from the 2005, 2010 and 2016 UAE Global School-Based Student Health Survey (GSHS)" revealed that the prevalence of inadequate physical activity was 77.1%, 77.5% and 79.7% respectively [16]. A meta-analysis that included 12,782 adolescents in UAE showed that almost a quarter of the adolescents have a sedentary lifestyle with no physical activity, less than half have been mildly involved in physical activities, and around a fifth practiced a moderate level of physical activity, and a quarter were involved in vigorous physical activity. Additionally, mild physical activity, was more common in female adolescents, whereas moderate and high physical activity, were significantly higher in male adolescents [17].

While the relationship between physical inactivity and mental health issues such as depression has been studied in international contexts [11], there is a clear lack of context-specific research focusing on adolescents in the United Arab Emirates (UAE). The UAE presents a unique sociocultural and environmental landscape, marked by rapid urbanization, increasing screen time, and a predominantly sedentary lifestyle, which may shape adolescent behaviors and mental health outcomes differently from those observed in other countries. Importantly, mental health has been identified as a national public health priority in the UAE [18]. Understanding how physical inactivity contributes to mental health challenges in this specific population is crucial for developing evidence-based, culturally appropriate interventions. Our study provides locally relevant data that can inform targeted strategies for schools, healthcare providers, and policymakers. In particular, the findings can support school authorities in establishing initiatives and protocols to address both depression and physical inactivity among students. Furthermore, the study helps raise awareness among adolescents and their parents, encouraging early prevention and promoting healthier behaviors both at home and in educational settings.

The present study aimed to assess the prevalence, factors, and relationship between physical inactivity and depression, and to determine the predictors of depression among adolescents in the UAE.

## Materials and methods

### Study design, sampling

A cross-sectional survey was conducted among school-going students in grades 9–12 at selected schools in Ajman Sharjah and Dubai. The study included adolescents from both genders and all nationalities whose parents signed Informed consent for participation. The study excluded adolescents who were not available at the data collection sites.

The sample size (n) was determined using the following equation: $n = Z^2 pq/d^2$ with a 95% confidence limit, adding 10% for a possible refusal rate. The proportion of physical inactivity among adolescents was calculated based on a previous study from the UAE, which showed an average rate of 79.7% [17]. The participants were selected based on convenience due to the logistical and practical constraints associated with accessing a representative sample of adolescents across schools in the UAE. The data collection was started in November 2022 and ended in April 2023. Analysis and manuscript finished by January 2024.

### Tools for data collection

Two standardized questionnaires were used to assess physical activity levels and depressive symptoms among adolescents

- The Physical Activity Questionnaire for Adolescents (PAQ-A) is a self-administered, 7-day recall instrument developed specifically for adolescents and has demonstrated good internal consistency in previous research [19]. In the current study, the PAQ-A demonstrated acceptable reliability, with a Cronbach's alpha of 0.779.

- The Centre for Epidemiologic Studies Depression Scale (CES-D) is a validated 20-item screening tool designed to measure depressive symptomatology in the general population, including adolescents. It has demonstrated strong psychometric properties and construct validity in a diverse population [20]. In our study, the Cronbach's α was 0.795, which indicates an acceptable reliability

In addition, information on socio-demography and factors related to physical activity and depression was obtained.

### Ethical issues

The study started after getting approval from GMU-IRB (Institutional Review Board) with reference number: IRB/COM/FAC/51/SEPT-2022, in accordance with the Declaration of Helsinki. Informed written consent was obtained from parents. Students were given the questionnaire. That was anonymous. We assured participants and their parents that the study was anonymous, the provided information was analysed groupwise, and there would be no link between the participant as a person and the results. Confidentiality of the information is respected, and only the research team and IRB Committee members may have access to the data. Data is stored for three years as per the university policy.

### Methodology

Final approval was obtained from the selected schools, and the research team had a meeting with the students to explain the study as well as have asked them to deliver informed consent to their parents to get their approval; students whose parents acknowledged their participation were included, and participants were handed the questionnaire to be filled. The cut-off level used in this study to identify physically active adolescents is > 2.73 according to Benítez-Porres et al. study [21]. The Crowding Index was employed as a proxy indicator of socioeconomic status. It was calculated by dividing the number of family members by the number of rooms in the household, explicitly excluding bathrooms, balconies, porches, foyers, hallways, and half-rooms, in line with the established definition [22]. According to this definition, crowding is considered present when there is more than one person per room, and severe crowding is defined as a Crowding Index greater than 1.5. This measure provides a contextually relevant and easily applicable indicator of household density, which can reflect socioeconomic constraints that may impact adolescent mental health.

### Data analysis

The data analysis was conducted using SPSS (Statistical Package for the Social Sciences) version 28. Both descriptive and inferential statistical methods were employed to summarize and interpret the findings. Descriptive statistics (e.g., means, standard deviations, and frequencies) were used to provide an overview of the dataset. A t-test was performed to compare mean differences between groups. A Correlation test was conducted to examine the relation between physical inactivity, depression, and other selected variables. To identify significant predictors of depression among adolescents, logistic regression analysis was applied, allowing for the evaluation of independent variables while controlling for potential confounders.

## Results

### Sociodemographic characteristics of participants

The study included 365 participants. Most participants were more than 14 years old (68.7%), females (60%), grade 10(32.1%), and their fathers' and mothers' education (college and postgraduate) were 83% and 87.7% respectively. 31% of fathers' jobs were Technicians & Associate professionals and 69.2% of the mother's jobs were Professional categories

according to the international classification of occupations. About one-third of adolescents (34.2%) were living in severely crowded houses, About 60% of the participants were the first order among their siblings, as shown in Table 1.

Around one-third (33.4%) of adolescents have a history of being diagnosed with COVID-19 infection. The history of other chronic diseases was low (Table 2).

Fig 1 depicts the prevalence of physical inactivity among participants. More than two-thirds of participants (68.8%) were physically inactive. Fig 2. Shows the prevalence of physical inactivity by gender and grades. The rate of physical inactivity is higher among senior students.

**Table 1. Sociodemographic characteristics of participants.**

| Variable | Subcategories | Number | % |
|---|---|---|---|
| Age | =<13 | 36 | 9.9 |
| | 14 | 78 | 21.35 |
| | 15 | 115 | 31.5 |
| | 16 | 67 | 18.35 |
| | =>17 | 69 | 18.9 |
| Gender | Male | 146 | 40.0 |
| | Female | 219 | 60.0 |
| Grade | 9 | 96 | 26.3 |
| | 10 | 117 | 32.1 |
| | 11 | 68 | 18.6 |
| | 12 | 84 | 23.0 |
| Father's Education Level | =<Secondary | 62 | 17.0 |
| | College | 157 | 43.0 |
| | Postgraduate | 146 | 40.0 |
| Mother's Education Level | =<Secondary | 45 | 12.3 |
| | College | 152 | 41.7 |
| | Postgraduate | 168 | 46.0 |
| Father Occupation (n=355) | Manager | 65 | 18.2 |
| | Professional | 56 | 15.8 |
| | Technicians & Associate professional | 111 | 31.3 |
| | Clerical Support workers | 45 | 12.7 |
| | Other | 78 | 22.0 |
| Mother Occupation (n=159) | Manager | 20 | 12.6 |
| | Professional | 110 | 69.2 |
| | Technicians & Associate professional | 10 | 6.3 |
| | Clerical Support workers | 16 | 10.0 |
| | Other | 3 | 1.9 |
| Crowding Index | Severe Crowding (>1.5) | 125 | 34.2 |
| | Not severe crowding | 240 | 65.8 |
| Order in Sibling | 1st | 216 | 59.7 |
| | 2nd | 109 | 30.1 |
| | 3rd | 22 | 6.1 |
| | > 3rd | 15 | 4.1 |

**Table 2. Medical history of participants.**

| Disease | No | | Yes | |
|---|---|---|---|---|
| | Number | % | Number | % |
| Covid 19 | 243 | 66.6 | 122 | 33.4 |
| Asthma | 348 | 95.3 | 17 | 4.7 |
| Other Respiratory Problems | 345 | 94.5 | 20 | 5.5 |
| Diabetes | 362 | 99.2 | 3 | 0.8 |
| Heart Problem | 363 | 99.5 | 2 | 0.5 |
| Hypertension | 358 | 98.1 | 7 | 1.9 |
| Abnormal Lipid Levels | 364 | 99.7 | 1 | 0.3 |
| Muscle Diseases | 362 | 99.2 | 3 | 0.8 |
| Bone Diseases | 359 | 98.4 | 6 | 1.6 |
| Kidney Problem | 362 | 99.2 | 3 | 0.8 |
| Other Health Problems | 323 | 88.5 | 42 | 11.5 |

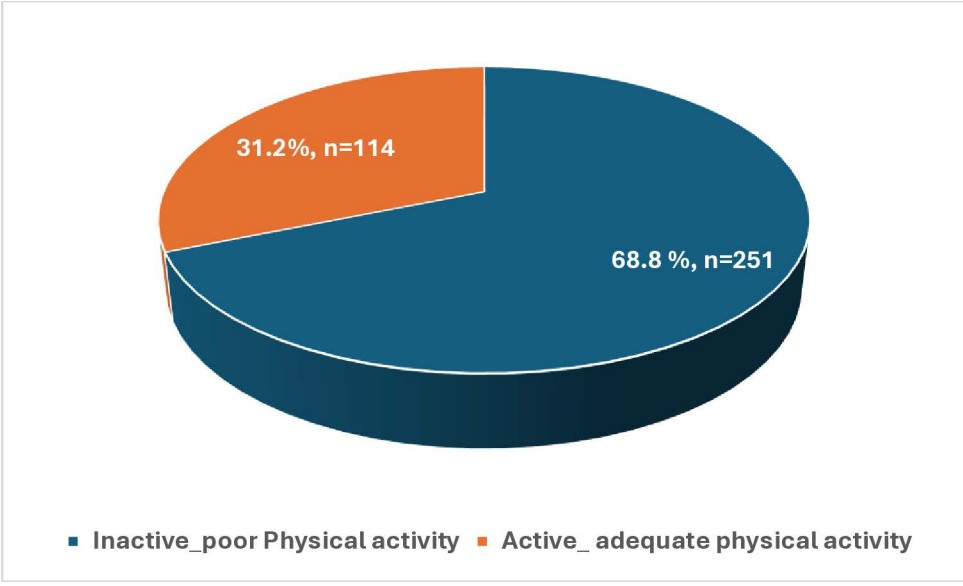

31.2%, n=114
68.8 %, n=251

■ Inactive_poor Physical activity  ■ Active_ adequate physical activity

**Fig 1. The prevalence of physical inactivity among participants.**

Analysis of the mean difference between physical activity and depression scores among different groups (Tables 3 and 4) showed significantly higher mean physical activity scores among males compared to females, p < 0.001 (Table 3), and significantly higher depression scores among females compared with males, p < 0.001 (Table 4). Other variables, such as the fathers' and mothers' education and occupation, showed non-significant differences concerning the mean PAQ score and CES-D score.

Tables 5 and 6 showed a significant negative correlation between physical activity and depression score CES-D, p < 0.001. Age showed a significant negative correlation to physical activity, p = 0.001 while showing an insignificant positive correlation with depression. Also, the grade was significantly negatively correlated to Physical activity, p = 0.005 while showed a significant positive correlation with depression, p = 0.027. Chronic health problems showed a significant negative

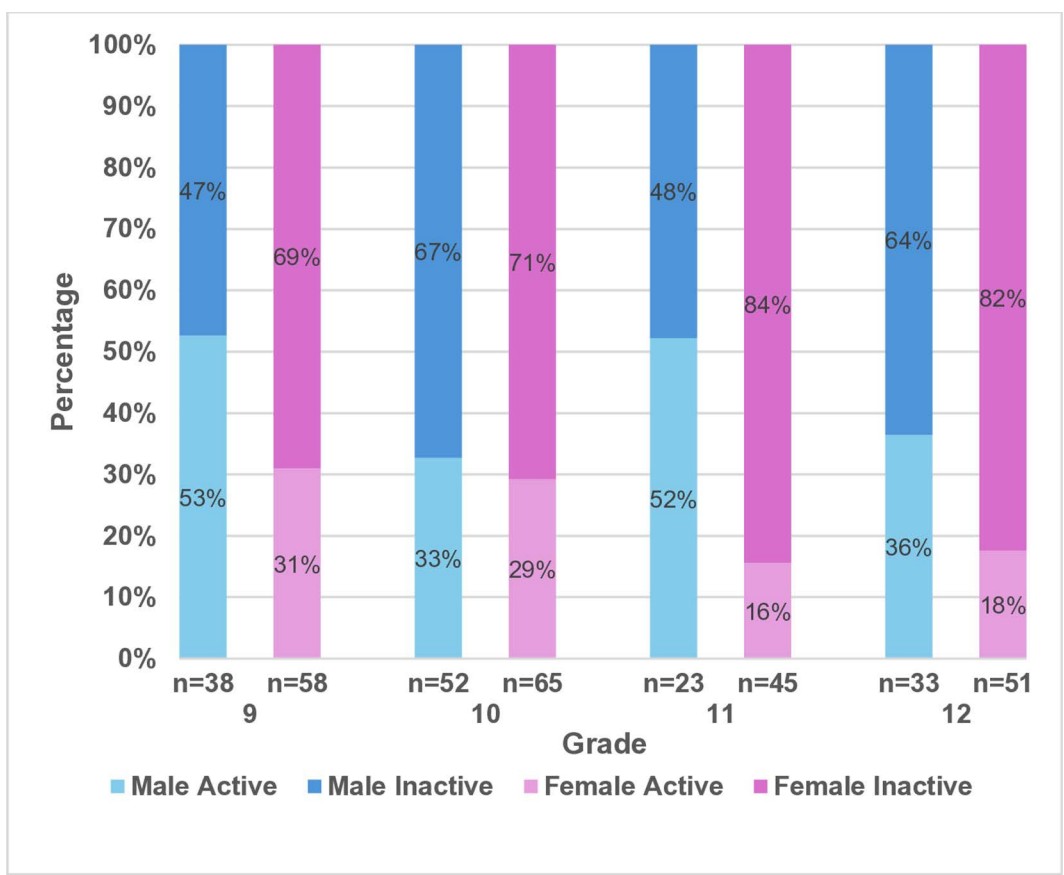

**Fig 2. The prevalence of physical activity by levels and grades among male and female adolescents.**

correlation with physical activity, p = 0.003, while showed a significant positive correlation with depression p < 0.001. Crowding index showed a significant negative correlation with depression, p = 0.036.

The prevalence of depression among participants was 51.2%, while the no depression was 48.8%, Fig 3.

Analysis of the mean difference in physical activity scores among depressed [2.28 ± 0.69] and nondepressed (2.62 ± 0.68) groups was significant (P < 0.001).

Fig 4. Showed that there is a negative and significant correlation between depression and physical activity scores, r = 0.29, P < 0.001. Table 7 shows the logistic regression analysis for predictors of depression among the studied participants.

In Table 7. The bivariable analysis showed that being females increase the likelihood of depression among adolescents by 98% compared to males. Also, senior students had a 95% higher likelihood of depression compared to junior students. Furthermore, students who were overweight or obese had a 3.2- and 4.4-times higher likelihood of depression compared to thin students, and inactive students had a 2.22-fold higher likelihood of depression compared to active students. In addition, adolescents who had reported having any chronic health problem had an increased likelihood of depression by 4.41 times compared to those without these health problems.

Further analysis of all significant variables in the multivariable logistic regression model revealed that being female had increased the likelihood of depression by 80%. Also, overweight and obese students had 4.34 times and 5.62 times, respectively, higher likelihood of depression compared to thin students. In addition, physically inactive adolescents had

**Table 3. The mean difference between physical activity scores among the studied participants.**

| Variable | Subcategories | Number | Mean PAQ Score | SD | Confidence Interval | | p |
|---|---|---|---|---|---|---|---|
| | | | | | Lower | Upper | |
| Gener | Male | 146 | 2.67 | 0.67 | 2.56 | 2.78 | <0.001 |
| | Female | 219 | 2.29 | 0.69 | 2.21 | 2.39 | |
| Father Education | =<Secondary | 62 | 2.49 | 0.66 | 2.33 | 2.66 | 0.305 |
| | College | 157 | 2.49 | 0.66 | 2.39 | 2.60 | |
| | Postgraduate | 146 | 2.38 | 0.76 | 2.25 | 2.50 | |
| Mother Education | =<Secondary | 45 | 2.43 | 0.72 | 2.21 | 2.65 | 0.969 |
| | College | 152 | 2.44 | 0.71 | 2.33 | 2.56 | |
| | Postgraduate | 168 | 2.46 | 0.70 | 2.35 | 2.56 | |
| Father Occupation | Manager | 65 | 2.51 | 0.66 | 2.35 | 2.67 | 0.846 |
| | Professional | 56 | 2.42 | 0.74 | 2.22 | 2.62 | |
| | Technicians & Associate Professional | 111 | 2.40 | 0.66 | 2.28 | 2.53 | |
| | Clerical Support Workers | 45 | 2.50 | 0.73 | 2.29 | 2.72 | |
| | Other | 78 | 2.46 | 0.73 | 2.29 | 2.62 | |
| Mother Occupation | Manager | 20 | 2.54 | 0.70 | 2.21 | 2.87 | 0.239 |
| | Professional | 110 | 2.50 | 0.71 | 2.37 | 2.63 | |
| | Technicians & Associate Professional | 10 | 2.05 | 0.48 | 1.71 | 2.39 | |
| | Clerical Support workers | 16 | 2.36 | 0.69 | 1.99 | 2.73 | |
| | Service and Sales Workers | 3 | 2.88 | 0.69 | 1.18 | 4.59 | |

**Table 4. The mean difference between depression scores among the studied participants.**

| Variable | Subcategories | Number | Mean CES-D Score | SD | Confidence Interval | | p |
|---|---|---|---|---|---|---|---|
| | | | | | Lower | Upper | |
| Gender | Male | 146 | 15.75 | 10.34 | 14.06 | 17.44 | <0.001 |
| | Female | 219 | 20.25 | 12.56 | 18.58 | 21.92 | |
| Father Education | =<Secondary | 62 | 18.16 | 12.75 | 14.92 | 21.40 | 0.174 |
| | College | 157 | 17.28 | 11.46 | 15.47 | 19.09 | |
| | Postgraduate | 146 | 19.83 | 11.96 | 17.87 | 21.79 | |
| Mother Education | =<Secondary | 45 | 18.44 | 12.90 | 14.57 | 22.32 | 0.468 |
| | College | 152 | 17.59 | 11.70 | 15.71 | 19.46 | |
| | Postgraduate | 168 | 19.23 | 11.85 | 17.43 | 21.04 | |
| Father Occupation | Manager | 65 | 20.63 | 13.76 | 17.22 | 24.04 | 0.275 |
| | Professional | 56 | 16.30 | 9.86 | 13.66 | 18.94 | |
| | Technicians & Associate Professional | 111 | 18.62 | 12.02 | 16.36 | 20.88 | |
| | Clerical Support Workers | 45 | 16.64 | 11.11 | 13.30 | 19.98 | |
| | Other | 78 | 19.00 | 12.28 | 16.23 | 21.77 | |
| Mother Occupation | Manager | 20 | 18.85 | 9.80 | 23.44 | 3.00 | 0.620 |
| | Professional | 110 | 19.96 | 12.02 | 22.24 | 0.00 | |
| | Technicians & Associate Professional | 10 | 25.60 | 10.39 | 33.04 | 7.00 | |
| | Clerical Support Workers | 16 | 19.25 | 9.55 | 24.34 | 5.00 | |
| | Service and Sales Workers | 3 | 21.00 | 14.93 | 58.10 | 10.00 | |

**Table 5. The Correlation between physical activity score with social, demographic, economic, and health factors.**

| Variables | Correlation Coefficient Between PAQ Score and Variables (r) | p |
|---|---|---|
| CES-D | −0.30 | <0.001 |
| Age | −0.17 | 0.001 |
| Birth order | 0.04 | 0.434 |
| Crowding Index | 0.03 | 0.534 |
| Grade | −0.15 | 0.005 |
| Have chronic health problem | −0.15 | 0.003 |

**Table 6. The Correlation between depression scores with social, demographic, economic, and health factors.**

| Variables | Correlation Coefficient Between CES-D Score and Variables (r) | p |
|---|---|---|
| Age | 0.10 | 0.055 |
| Birth order | 0.02 | 0.727 |
| Crowding Index | −0.11 | 0.036 |
| Grade | 0.12 | 0.027 |
| Have chronic health problem | 0.37 | <0.001 |

a higher likelihood of depression by 92% compared to physically active adolescents after adjusting for other variables in the model. Also, students with chronic health problems had a 4.56-fold higher likelihood of depression compared to their peers who had not reported having these problems.

## Discussion

Physical activity has many health benefits for young people. This study showed that 31.2% of adolescents are physically inactive. These results may be due to the sedentary lifestyle among adolescents in the UAE. Our finding agrees with a study conducted by Guthold et al., who reported that more than 80% of adolescents aged 11–17 years did not meet their daily recommended physical activity [23].

Our results also showed that about half of the participants had symptoms of depression. The prevalence of depression among participants was 51.2%. This result is considered alarmingly high compared to previous studies about depression [24,25]. A study conducted in Malaysia, Kaur et al., found that 17.7% of the participants had depressive symptoms [24]. Shorey et al reported that the Global Point prevalence of elevated depressive symptoms among adolescents increased from 24% between 2001 and 2010 to 37% between 2011 and 2020, and the highest prevalence of elevated depressive symptoms was found in the Middle East, Africa, and Asia [25]. The high prevalence of depression in our research could be due to challenges the students face related to their studies, such as lack of sleep, the burden of assignments and exams, and concerns about the GPA score [26]. Also, according to a study that was conducted at the end of the COVID-19 Pandemic; the lockdown and lack of social activity were related to the high rate of depression among adolescents [14].

This study showed a negative and significant correlation between physical activity and depressive symptoms. Also, we found that physically inactive adolescents had a 92% higher likelihood of depression compared to physically active adolescents. This finding agrees with the meta-analysis of prospective cohort studies. The previous study found that people with high levels of physical activity had lower odds of developing depression [Adjusted Odd Ratio = 0.83,

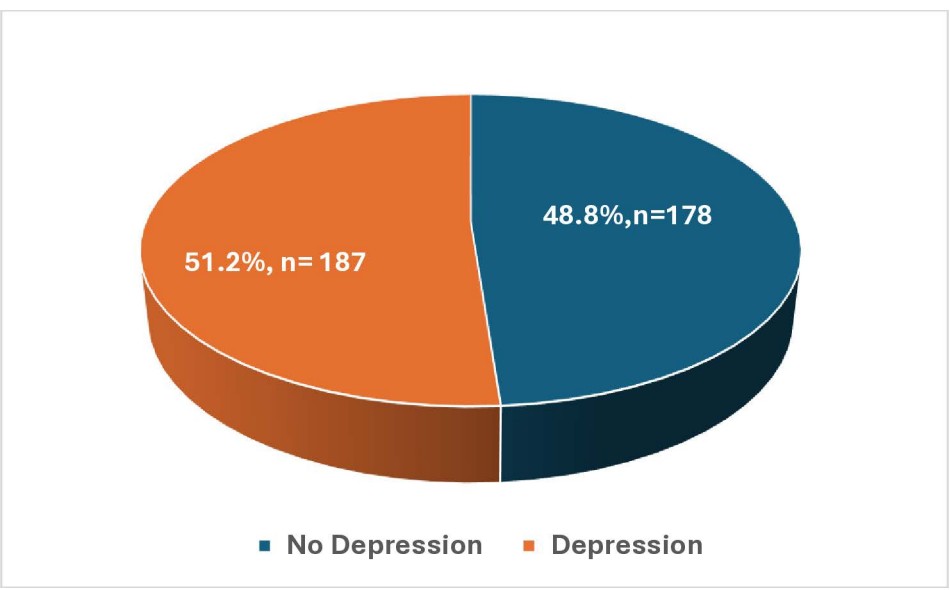

**Fig 3. The prevalence of depression among adolescents.**

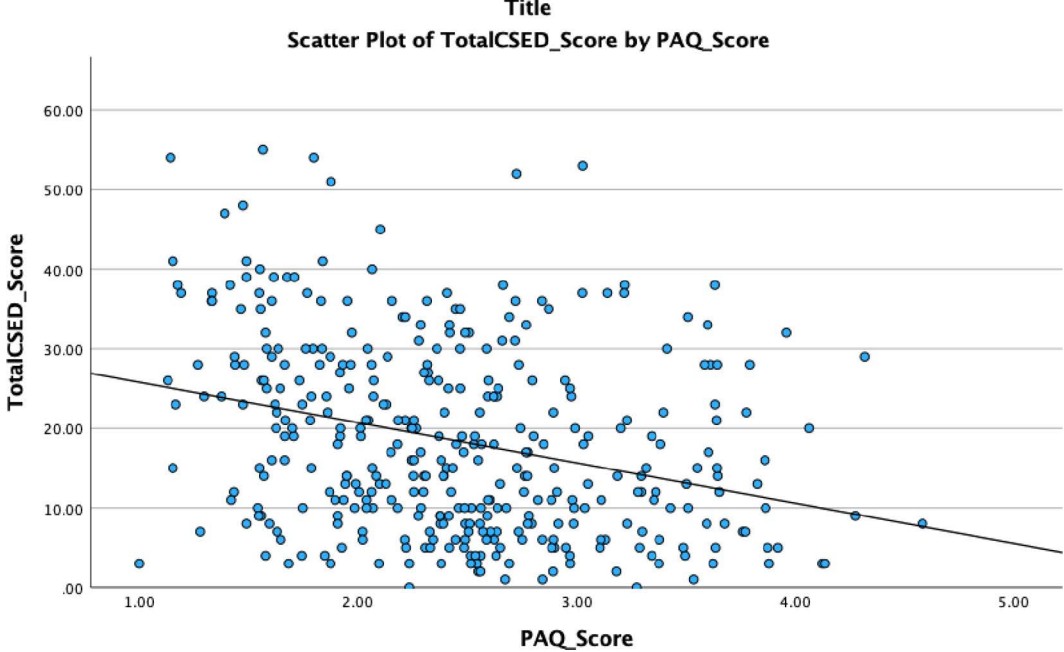

**Fig 4. Scatter diagram for the correlation between the physical activity scores and the depression scale scores.**

95%CI:0.79–0.88]. The authors concluded that physical activity has a protective effect against the emergence of depression in youth [27]. It is shown that exercise increases feelings of control, independence, feelings of well-being, and confidence [28].

**Table 7. Logistic regression analysis for predictors of depression among adolescents.**

| Variable | Subcategory | COR (95% CI) | p-value | AOR (95% CI) | p-value |
|---|---|---|---|---|---|
| Age (Year) | =<13 | 1.00 (Ref) | — | — | — |
|  | 14 | 0.97 (0.44–2.14) | 0.932 | — | — |
|  | 15 | 1.57 (0.74–3.33) | 0.241 | — | — |
|  | 16 | 1.01 (0.45–2.29) | 0.974 | — | — |
|  | =>17 | 2.07 (0.91–4.69) | 0.082 | — | — |
| Gener | Male | 1.00 (Ref) |  | — | — |
|  | Female | 1.98 (1.29–3.03) | **0.002** | 1.80 (1.11–2.90) | 0.017 |
| Father Education | =<Secondary | 1.00 (Ref) |  | — | — |
|  | College | 0.95 (0.53–1.71) | 0.867 | — | — |
|  | Postgraduate | 1.41 (0.77–2.55) | 0.263 | — | — |
| Mother Education | =<Secondary | 1.00 (Ref) |  | — | — |
|  | College | 1.07 (0.55–2.09) | 0.835 | — | — |
|  | Postgraduate | 1.15 (0.60–2.22) | 0.677 | — | — |
| Father Occupation | Manager | 1.13 (0.58–2.20) | 0.713 | — | — |
|  | Professional | 0.74 (0.37–1.48) | 0.397 | — | — |
|  | Technicians & Associate professional | 0.87 (0.49–1.56) | 0.646 | — | — |
|  | Clerical Support Workers | 0.75 (0.36–1.57) | 0.443 | — | — |
|  | Other | 1.00 (Ref) |  | — | — |
| Mother Occupation | Manager | 3.00 (0.23–38.88) | 0.401 | — | — |
|  | Professional | 2.68 (0.24–30.45) | 0.426 | — | — |
|  | Technicians & Associate professional | 18.00 (0.76–427.29) | 0.074 | — | — |
|  | Clerical Support Workers | 2.57 (0.19–34.47) | 0.476 | — | — |
|  | Other | 1.00 (Ref) |  | — | — |
| Birth Order in Sibling | 1st | 1.00 (Ref) |  | — | — |
|  | 2nd | 1.06 (0.67–1.68) | 0.814 | — | — |
|  | 3rd | 1.82 (0.73–4.51) | 0.198 | — | — |
|  | >3rd | 2.08 (0.69–6.27) | 0.196 | — | — |
| Grade | 9 | 1.00 (Ref) |  | 1.00 (Ref) | — |
|  | 10 | 1.71 (0.99–2.94) | 0.055 | 1.59 (0.87–2.92) | 0.135 |
|  | 11 | 1.74 (0.93–3.27) | 0.082 | 1.05 (0.51–2.14) | 0.904 |
|  | 12 | 1.95 (1.08–3.53) | 0.028 | 1.40 (0.72–2.71) | 0.325 |
| Crowding Index | Severe Crowding | 1.10 (0.71–1.70) | 0.666 | — | — |
|  | Not Severe Crowding | 1.00 (Ref) | — | — | — |
| Have any Chronic Health problem | No | 1.00 (Ref) | — | 1.00 (Ref) | — |
|  | Yes | 4.41 (2.61–7.48) | <0.001 | 4.56 (2.57–8.08) | <0.001 |
| BMI for Age | Thin | 1.00 (Ref) |  | 1.00 (Ref) | — |
|  | Normal | 2.14 (0.72–6.37) | 0.171 | 2.57 (0.74–8.94) | 0.136 |
|  | Overweight | 3.20 (1.02–10.07) | 0.047 | 4.34 (1.18–15.99) | 0.028 |
|  | Obese | 4.40 (1.20–16.17) | 0.026 | 5.62 (1.28–24.73) | 0.023 |
| PAQ Activity Level (PAQ_A) | Inactive | 2.22 (1.41–3.50) | <0.001 | 1.92 (1.16–3.19) | 0.012 |
|  | Active | 1.00 (Ref) | — | 1.00 (Ref) | — |

COR = Crude Odds Ratio; AOR = Adjusted Odds Ratio; CI = Confidence Interval.

Physical activity produces its antidepressant effect through multiple biological and psychosocial pathways. Physical exercise enhances insulin-like growth factor (IGF)-1 and activates PGC-1α/FNDC5/Irisin pathway. Physical exercise also increases the expression of brain-derived neurotrophic factor (BDNF) and its receptor in the hippocampus and prefrontal cortex, leading to the inhibition of depressive-like behavior [29]. Moreover, physical activity increases neurotrophins like vascular endothelial growth factor (VEGF), which stimulate angiogenesis, causing lasting changes in brain structure that improve its vasculature and brain functioning [30]. A study conducted by Hamedinia et al. reported that eight weeks of aerobic training significantly increased serotonin levels, and eight weeks of aerobic and anaerobic exercise significantly increased BDNF [31].

Our findings suggest a strong link between physical inactivity, obesity, and depression among adolescents. While the primary objective of this study was to examine the relationship between physical inactivity and depression, the inclusion of obesity and chronic conditions such as diabetes and hypertension was essential, as these variables may act as potential confounders or mediators in this association. Obesity and chronic diseases have been associated with both lower physical activity levels and increased risk of depressive symptoms [32–34]. By accounting for these factors, our analysis provides a deeper understanding of the complex interplay between physical and mental health. This study indicates that physically inactive adolescents had a 92% higher likelihood of experiencing depression, after adjusting for obesity and chronic diseases. This underscores the independent protective role of physical activity in promoting mental well-being, regardless of existing physical health conditions. Given that adolescents with obesity or chronic diseases are already at elevated risk for depressive symptoms, as was seen in our study and other researchers' findings [32–35], engaging in regular physical activity may serve as a critical intervention to buffer against these mental health challenges. Physical activity contributes not only to improved physical health but also to enhanced mood, reduced stress, and better emotional regulation, mechanisms that are particularly relevant for youth managing chronic conditions. Therefore, integrating physical activity into daily routines may help adolescents with obesity or chronic illnesses better manage their overall health and reduce the likelihood of developing or worsening depressive symptoms. This is documented in a previous systematic review among patients with chronic diseases [36].

The latter suggestion is also supported by a meta-analysis for the effects of physical activity on depression, anxiety, and weight-related outcomes among children and adolescents with overweight/obesity. The study showed that physical activity combined with other interventions has a significant effect both on anxiety symptoms and BMI [37].

Sociodemographic factors such as gender are related to both physical activity and depression. Adolescent girls tend to have lower physical activity levels and higher rates of depression compared to boys [38]. In the same vein, a study conducted in South Korea by Kim et al. reported that the prevalence of depression was higher in females than males and that the depression level increased with age [39], which is consistent with the age and gender disparity observed in our study. A similar trend was also reported in other studies; for instance, a cross-sectional study among secondary school children in Dhaka City, Bangladesh, reported depression rates of 30% in females and 19% in males [40]. Similarly, Mridha et al found that the prevalence of depression for females and males was 27% and 22% respectively [6]. Bhattarai et al further highlighted that students who had low perceived social support, those who did not share their problems with anyone, and had low self-esteem were at higher risk of being depressed [8]. Accortt et al. linked the higher prevalence of depression among females with social, psychological, and biological variables. The authors suggested that females are more liable to depression due to hormonal factors during the menstrual cycle, low self-esteem, and a history of trauma or abuse [41].

Numerous studies, like those cited before, consistently reported higher rates of depression among adolescent females compared to their male counterparts, suggesting potential biological, social, and psychological influences. However, limited research has explored the role of physical activity in this gender disparity. Since physical activity is known to enhance mood, regulate stress hormones, and support overall well-being, our findings suggest that differences in activity levels may partly explain the higher prevalence of depression among females. Our findings on gender disparities in adolescent depression within the sociocultural context of the UAE underscore the urgent need for gender-sensitive mental health

strategies. The significantly higher risk of depression among female adolescents highlights their heightened vulnerability and calls for culturally tailored interventions that address stigma, improve access to care, and promote protective factors such as physical activity. Targeted programs in schools and communities that encourage physical engagement may serve as effective strategies to enhance mental well-being and reduce inequities in mental health outcomes.

In this study, we examined parental education levels and employment status as key socioeconomic determinants potentially influencing adolescent physical activity and depression. However, our analysis did not reveal any statistically significant associations between these variables and the outcomes of interest. These findings suggest that, within our study population, these specific aspects of socioeconomic status may not have played a determining role in shaping adolescents' physical activity patterns or mental health status. This contrasts with findings from other contexts, such as a study conducted in China, which reported significant positive correlations between parental education and adolescent physical activity (mother: r = 0.798; father: r = 0.793), as well as between parental occupation and physical activity (mother: r = 0.549; father: r = 0.479) [42]. The discrepancy may be attributable to cultural, economic, or contextual differences across populations, emphasizing the importance of localized research to understand how socioeconomic factors operate within specific settings."

Low physical activity among adolescents in the UAE has been linked to other sociocultural, behavioural, and environmental factors. For instance, Henry et al. [43] reported that female adolescents in the UAE had particularly low activity levels due to weather conditions, cultural restrictions, and unsupportive community attitudes. They also found high leisure-time sedentary behaviour in boys (51.1%) and girls (66.7%), well above the global average (26.4%) and the highest among 10 Eastern Mediterranean countries [44].

Housing is a well-known socioeconomic determinant of health. It has been documented that household overcrowding negatively affects physical and mental health, but decreasing overcrowding is not associated with a significant decrease in depression [45].

Our findings indicate a significant negative correlation between the crowding index and depression, suggesting that higher household density may be associated with lower depressive symptoms. However, the lack of a significant relationship in regression analysis indicates that this correlation may be influenced by other confounding factors rather than a direct link. It is possible that family size may act as a moderating factor against stress and depression in certain contexts. For example, Ugwu et al. [46] found that perceived family cohesion and larger family size moderated the relationship between burnout and recovery among intensive care unit doctors in Southeastern Nigeria. However, this finding may be culturally specific and not fully applicable to the UAE, given that the UAE has undergone rapid socio-economic transformation in recent decades, which has affected traditional family dynamics. Young people increasingly rely on the Internet and social media, leading to reduced affectionate face-to-face interactions and less quality time with family members [47]. As a result, even in larger households, the protective effect of family cohesion may be weaker, potentially limiting its ability to counteract the negative impact of crowding on mental health. Further research is needed to explore the complex interactions between household environments, social dynamics, and mental health outcomes.

It's worth mentioning that although our study did not directly measure social media use, it is important to contextualize our findings within the broader literature on adolescent mental health. Increasing evidence suggests that excessive or problematic social media use may be associated with a heightened risk of depression among adolescents. For instance, a cross-sectional study conducted in Nigeria found that 56.5% of adolescents demonstrated poor mental health outcomes linked to high levels of social media engagement [48]. Mechanisms proposed in the literature include social comparison, cyberbullying, disrupted sleep patterns, and reduced face-to-face interactions, all of which can exacerbate feelings of isolation, anxiety, and depressive symptoms. While our findings align with the general concern around adolescent mental health, they highlight the need for future research that specifically investigates the mediating or moderating role of social media use in the context of physical inactivity and mental well-being.

This study has some limitations. First, a convenient sampling method was used to recruit participants, and the study was limited to adolescents from selected schools in the Emirates of Ajman, Sharjah, and Dubai, based on logistical feasibility, accessibility, and school-level approvals. This limits the generalizability of the findings. However, although not nationally representative, these Emirates encompass diverse and densely populated areas, offering meaningful insights into adolescent mental health and physical activity patterns in the studied contexts. Additionally, while we examined several relevant factors associated with depression and physical inactivity, other contextual or psychosocial influences may have been overlooked.

Also, being a cross-sectional study, this design precludes the ability to establish a temporal relationship between physical activity and depressive symptoms; therefore, we cannot be sure whether lower depressive symptoms were a result of higher physical activity

## Conclusion

Inactivity and depressive symptoms are highly prevalent among adolescents in the UAE, with only 31.2% meeting the recommended levels of physical activity, and 51.2% of them reported depressive symptoms. Predictors of depression are physical inactivity, female gender, overweight and obesity, and having a chronic disease.

Our findings suggest that higher physical activity levels may help mitigate depressive symptoms. These results underscore the vital role of physical activity in supporting adolescent mental well-being and emphasize the need for targeted interventions to promote an active lifestyle as a protective factor against depression.

## Recommendations

The research revealed a critical need for interventions aimed at promoting physical activity as a potential strategy to alleviate depressive symptoms among UAE adolescents by:

1. Public health campaigns to increase public and parental awareness about the importance of physical activity and the mental well-being of their children and encourage active lifestyles.

2. Implement active school-based and community-based sporting initiatives fitting UAE's climate and culture. These initiatives should be complemented with accessible school-based support services, such as counseling and mental health education, to promote early identification and management of depressive symptoms.

3. A future nationally representative longitudinal studies that capture regional differences and provide a more comprehensive understanding of the factors shaping adolescent mental well-being and physical activity from the adolescents' perspectives

## Supporting information

**S1 File. Supporting information.**
(PDF)

## Author contributions

**Conceptualization:** Faten Mahmoud Diab, Shatha AL-Sharbatti.

**Data curation:** Ghada Elsayed Elgarawany.

**Formal analysis:** Shatha AL-Sharbatti.

**Investigation:** Shatha AL-Sharbatti, Abdalla Tamer Eltanbadawy, Rami Aws Alfahad, Ghada Elsayed Elgarawany.

**Methodology:** Shatha AL-Sharbatti, Abdalla Tamer Eltanbadawy, Rami Aws Alfahad.

**Project administration:** Shatha AL-Sharbatti, Ghada Elsayed Elgarawany.

**Resources:** Shatha AL-Sharbatti, Abdalla Tamer Eltanbadawy, Rami Aws Alfahad.

**Supervision:** Faten Mahmoud Diab, Shatha AL-Sharbatti.

**Validation:** Faten Mahmoud Diab, Shatha AL-Sharbatti, Ghada Elsayed Elgarawany.

**Visualization:** Faten Mahmoud Diab, Shatha AL-Sharbatti, Ghada Elsayed Elgarawany.

**Writing – original draft:** Shatha AL-Sharbatti, Abdalla Tamer Eltanbadawy, Rami Aws Alfahad, Ghada Elsayed Elgarawany.

**Writing – review & editing:** Faten Mahmoud Diab, Shatha AL-Sharbatti, Ghada Elsayed Elgarawany.

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
