## [Decision Letter · Decision Letter 0]

15 Jul 2025

Dear Dr. Elgarawany,

Thank you for submitting your manuscript to PLOS ONE. After careful consideration, we feel that it has merit but does not fully meet PLOS ONE’s publication criteria as it currently stands. Therefore, we invite you to submit a revised version of the manuscript that addresses the points raised during the review process.

**ACADEMIC EDITOR: ** Dear authors, please address all the reviewers’ comments and make necessary corrections to the manuscript, in order to improve its clarity, scientific rigor, and overall quality

We look forward to receiving your revised manuscript.

Kind regards,

Zulkarnain Jaafar

Academic Editor

PLOS ONE

Journal Requirements:

Reviewers' comments:

Reviewer's Responses to Questions

**Comments to the Author**

1. Is the manuscript technically sound, and do the data support the conclusions?

Reviewer #1: Partly

Reviewer #2: Yes

2. Has the statistical analysis been performed appropriately and rigorously?

Reviewer #1: Yes

Reviewer #2: Yes

3. Have the authors made all data underlying the findings in their manuscript fully available?

Reviewer #1: Yes

Reviewer #2: Yes

4. Is the manuscript presented in an intelligible fashion and written in standard English?

Reviewer #1: Yes

Reviewer #2: Yes

Reviewer #1: Dear Authors

I have read your manuscript in which you have investigated the Impact of Physical Inactivitye on Mental Health of UAE Adolescents. The paper is well written and provides important information; Nevertheless, there are some considerations need to be addressed here.

- This study aims to examine the relationship between variables (e.g., physical activity and depression) whose associations have already been well-documented and established in prior literature. The novelty of the research topic has not been sufficiently justified.

- Similar ambiguity exists regarding other research questions, such as the relationship between gender and depression.

- What was the total population, and why was only a subset from 3 part of the country in the study? A clear rationale for this selection is necessary.

- The study does not provide sufficient details on the validity and reliability of the questionnaires used.

- The results section reports on parental education levels, but this data has not been adequately discussed in the discussion section. The same applies to parental employment status, which was mentioned but not thoroughly analyzed.

- Although the study acknowledges the influence of social media on adolescent depression, this aspect is not explored in the discussion.

- If the primary focus is on the relationship between physical inactivity and depression, the inclusion of secondary outcomes (such as obesity or chronic diseases linked to inactivity) seems unnecessary unless directly tied to the study’s main objectives.

Reviewer #2: In my opinion, the topic is extremely pertinent and speaks about a worldwide public health crisis (physical inactivity and adolescent mental well-being) with a special emphasis on the UAE, with sparse data present. The manuscript points to a startling prevalence of depression and physical inactivity among adolescents in the UAE, highlighting the urgency for interventions. The post-COVID-19 context of the study lends timeliness, given that the pandemic had amplified mental health issues. The manuscript is well-articulated in expressing its purpose: to evaluate the prevalence, determinants, and association of physical inactivity with depression, and predictors of depression. The organization is in a typical order (Introduction, Methods, Results, Discussion, Conclusion), which makes it easy to read and follows journal requirements.

There are some grammatical, typographical, and formatting errors in the manuscript. For instance, inconsistent usage of terms (e.g., "Crowding Index" vs. "American Classification of Crowing Index"), stilted wording (e.g., "human evolution process" in line 77), and omitted punctuation in tables (e.g., Table 7). These detract from professionalism and readability. Some sentences are too long or ambiguous, like the explanation of the crowding index (lines 419–429), which is unclear in describing the reverse, negative correlation with depression. The text does sometimes employ old-fashioned or inexact vocabulary (e.g., "mental problems" rather than "mental health disorders" in line 79).

The introduction is a strong backdrop, drawing on international and local statistics regarding physical inactivity and mental health, which places the study well.

Under methodology, it will be a good idea to explain why convenience sampling is being used and present its weaknesses in the discussion section. Include questionnaire administration details (e.g., language, mode of delivery) and ensure that PAQ-A and CES-D have cultural validity in the UAE context. It would be advisable to clearly state the limitation of the cross-sectional design in establishing causality and not use causal words in the discussion unless such can be confirmed with longitudinal data.

The application of descriptive and inferential statistics (t-tests, correlation, logistic regression) is fitting to the research questions, and the justification of sample size calculation using available UAE data is valid. The finding of the strong negative correlation between physical activity and depression (p<0.05) and the identification of predictors such as female gender, obesity, chronic disease are interesting findings. The multivariable logistic regression adds support to the results by adjusting for confounders, providing added depth to the analysis.

Author could consider, condense the results section by eliminating duplicative language and making sure all statistical results (e.g., correlation coefficients, effect sizes) are adequately reported with confidence intervals. Consider including a chart that graphically displays the most important findings, like the prevalence of physical activity or depression by gender or grade, to make readability easier. The discussion effectively contrasts the results with other literature, referring to studies in diverse regions (e.g., Malaysia, South Korea, Bangladesh) and meta-analyses as evidence supporting the findings' validity.

The paper suggests biological and psychosocial processes (e.g., BDNF, IGF-1, serotonin) connecting physical activity with decreased depression, exhibiting a careful synthesis of inter-disciplinary evidence. Practical suggestions (e.g., active school promotion, public health campaigns) are policy-relevant and implementable for policymakers and educators. Explain contradictory results by mentioning possible methodological variances or contextual conditions. Investigate UAE-specific conditions (e.g., cultural beliefs toward physical activity, gender segregation in schools, or environmental issues such as heat) that could affect the findings.

Offer a stronger explanation for the crowding index result, perhaps incorporating social support or socioeconomic literature in the UAE.

Provide actionable interventions, for example, implementing physical education in school curricula, providing gender-specific programs to tackle female inactivity, or establishing community-based sporting initiatives fitting UAE's climate and culture.

Provide specific research gaps, for example, longitudinal studies or studies into cultural barriers to physical activity. Revise and resubmit after having responded to the identified weaknesses, notably clarity, methodological transparency, and UAE-specific contextualization.

**Do you want your identity to be public for this peer review?** For information about this choice, including consent withdrawal, please see our Privacy Policy

Reviewer #1: No

Reviewer #2: **Yes: ** Nitesh Bansal

---

## [Author Response · Author response to Decision Letter 1]

6 Aug 2025

Responses to Reviewers

Respected reviewer 1. Thank you very much for reviewing our manuscript and for the constructive comments and suggestions you provided to improve the quality of the publication

Comment one: Reviewer 1: This study aims to examine the relationship between variables (e.g., physical activity and depression) whose associations have already been well-documented and established in prior literature. The novelty of the research topic

has not been sufficiently justified.

Response of Authors: Thank you for this comment. The following statement is added as a rationale to explain the relevance of the topic within the context of UAE adolescents (lines 116-129 in the revised manuscript).

“While the relationship between physical inactivity and mental health issues such as depression has been studied in international contexts, there is a clear lack of context-specific research focusing on adolescents in the United Arab Emirates (UAE). The UAE presents a unique sociocultural and environmental landscape, marked by rapid urbanization, increasing screen time, and a predominantly sedentary lifestyle, which may shape adolescent behaviors and mental health outcomes differently from those observed in other countries.

Importantly, mental health has been identified as a national public health priority in the UAE [18]. Understanding how physical inactivity contributes to mental health challenges in this specific population is crucial for developing evidence-based, culturally appropriate interventions. Our study provides locally relevant data that can inform targeted strategies for schools, healthcare providers, and policymakers.

In particular, the findings can support school authorities in establishing initiatives and protocols to address both depression and physical inactivity among students. Furthermore, the study helps raise awareness among adolescents and their parents, encouraging early prevention and promoting healthier behaviors both at home and in educational settings.

The reference below will be incorporated into the manuscript, with appropriate adjustments made to the in-text citation numbering.

18. Public Policy Document. The National Policy for the Promotion of Mental Health.

https://www.uaelegislation.gov.ae/en/policy/details/the-national-policy-for-the-promotion-of-mental-health

Comment two: Reviewer 1

Similar ambiguity exists regarding other research questions, such as the relationship between gender and depression

Response of Authors:

Although gender differences in depression have been previously observed in global studies, our findings contribute important context-specific evidence by highlighting that female adolescents in the UAE are significantly more likely to report depressive symptoms, with an adjusted odds ratio (AOR = 1.80), even after controlling for key confounders.

This association warrants particular attention within the UAE, where gender norms, academic pressures, social expectations, and cultural factors may differentially impact mental health in adolescent girls. Moreover, access to mental health support and help-seeking behavior can be influenced by gender-specific stigma and family dynamics in this region. Therefore, examining gender differences is not merely a repetition of global findings, but a necessary step toward informing gender-sensitive mental health strategies and targeted school-based interventions that are culturally appropriate for the UAE context.

The following statements are added to the discussion to reflect on this finding [lines 366-373].

“Our findings on gender disparities in adolescent depression within the sociocultural context of the UAE underscore the urgent need for gender-sensitive mental health strategies. The significantly higher risk of depression among female adolescents highlights their heightened vulnerability and calls for culturally tailored interventions that address stigma, improve access to care, and promote protective factors such as physical activity. Targeted programs in schools and communities that encourage physical engagement may serve as effective strategies to enhance mental well-being and reduce inequities in mental health outcomes”.

Comment Three: Reviewer 1

What was the total population, and why was only a subset from 3 part of the country in the study? A clear rationale for this selection is necessary.

Response of Authors:

The study sample was drawn from selected schools in Ajman, Sharjah, and Dubai due to logistical feasibility, population accessibility, and school-level approvals. A nationwide study including all seven emirates was beyond the scope of the current project due to resource constraints, time limitations, and the administrative complexity of securing approvals across multiple educational zones. Nonetheless, we think that the selected sample provides meaningful insights into adolescent mental health and physical activity patterns.

This statement is added in the Study design, sampling[ lines: 142-144]“The participants were selected based on convenience due to the logistical and practical constraints associated with accessing a representative sample of adolescents across schools in the UAE”.

Also, this statement will be added as a limitation in the discussion [line 423-430]

“This study was limited to adolescents from selected schools in the Emirates of Ajman, Sharjah, and Dubai, based on logistical feasibility, accessibility, and school-level approvals. Although not nationally representative, these Emirates encompass diverse and densely populated areas, offering meaningful insights into adolescent mental health and physical activity patterns in the studied contexts”.

Additionally, while we examined several relevant factors associated with depression and physical inactivity, other contextual or psychosocial influences may have been overlooked.

Comment Four: Reviewer 1

The study does not provide sufficient details on the validity and reliability of the questionnaires used.

Response of Authors:

Thank you for this comment.

The following statements are added in the “Tools for data collection”[lines:147-157]

“Two standardized questionnaires were used to assess physical activity levels and depressive symptoms among adolescents

-The Physical Activity Questionnaire for Adolescents (PAQ-A) is a self-administered, 7-day recall instrument developed specifically for adolescents and has demonstrated good internal consistency in previous research [19]. In the current study, the PAQ-A demonstrated acceptable reliability, with a Cronbach’s alpha of 0.779.

-The Centre for Epidemiologic Studies Depression Scale (CES-D) is a validated 20-item screening tool designed to measure depressive symptomatology in the general population, including adolescents. It has demonstrated strong psychometric properties and construct validity in a diverse population [20]. In our study, the Cronbach’s α was 0.795, which indicates an acceptable reliability

The references below will be incorporated into the manuscript, with appropriate adjustments made to the in-text citation numbering.

19. Martínez-Gómez D, Martínez-de-Haro V, Pozo T, Welk GJ, Villagra A, Calle ME, Marcos A, Veiga OL. Fiabilidad y validez del cuestionario de actividad física PAQ-A en adolescentes españoles [Reliability and validity of the PAQ-A questionnaire to assess physical activity in Spanish adolescents]. Rev Esp Salud Publica. 2009 May-Jun;83(3):427-39. Spanish. doi: 10.1590/s1135-57272009000300008. PMID: 19701574.

20. Radloff, L. S. The CES-D Scale: A Self-Report Depression Scale for Research in the General Population. Applied Psychological Measurement. 1977;1(3), 385-401. https://doi.org/10.1177/014662167700100306

Comment five: Reviewer 1

The results section reports on parental education levels, but this data has not been adequately discussed in the discussion section. The same applies to parental employment status, which was mentioned but not thoroughly analyzed

Response of Authors:

Thank you for this valuable comment. We acknowledge that the manuscript did not adequately address the findings related to parental education levels and employment status in the discussion section. We have now added the following paragraph to strengthen the discussion: [lines:374-385]

"In this study, we examined parental education levels and employment status as key socioeconomic determinants potentially influencing adolescent physical activity and depression. However, our analysis did not reveal any statistically significant associations between these variables and the outcomes of interest. These findings suggest that, within our study population, these specific aspects of socioeconomic status may not have played a determining role in shaping adolescents' physical activity patterns or mental health status. This contrasts with findings from other contexts, such as a study conducted in China, which reported significant positive correlations between parental education and adolescent physical activity (mother: r = 0.798; father: r = 0.793), as well as between parental occupation and physical activity (mother: r = 0.549; father: r = 0.479) [42]. The discrepancy may be attributable to cultural, economic, or contextual differences across populations, emphasizing the importance of localized research to understand how socioeconomic factors operate within specific settings."

The reference below will be incorporated into the manuscript, with appropriate adjustments made to the in-text citation numbering.

42. Yang W, Xiang Z, Hu H, Zheng H and Zhao X. The impact of family

socioeconomic status on adolescent mental and physical health: the mediating role of

parental involvement in youth sports. Front. Public Health 2025;13:1540968.

doi: 10.3389/fpubh.2025.1540968

Comment Six: Reviewer 1

Although the study acknowledges the influence of social media on adolescent depression, this aspect is not explored in the discussion.

Response of Authors:

Thank you for this insightful observation. We agree that social media use is a significant factor in adolescent mental health and warrants further discussion. In response, we have revised the discussion section to include a paragraph that explores the relationship between social media use and depression among adolescents.[in discussion, lines:411-422]

"Although our study did not directly measure social media use, it is important to contextualize our findings within the broader literature on adolescent mental health. Increasing evidence suggests that excessive or problematic social media use may be associated with a heightened risk of depression among adolescents. For instance, a cross-sectional study conducted in Nigeria found that 56.5% of adolescents demonstrated poor mental health outcomes linked to high levels of social media engagement [48]. Mechanisms proposed in the literature include social comparison, cyberbullying, disrupted sleep patterns, and reduced face-to-face interactions, all of which can exacerbate feelings of isolation, anxiety, and depressive symptoms. While our findings align with the general concern around adolescent mental health, they highlight the need for future research that specifically investigates the mediating or moderating role of social media use in the context of physical inactivity and mental well-being."

The reference below will be incorporated into the manuscript, with appropriate adjustments made to the in-text citation numbering.

Ref:48. Anche DA, Akafa TA, Karimu S, Akafa VT, Oladele GO. From Connection to Concern: Understanding Social Media's Influence on Mental Health Among Adolescents in

Abuja, Nigeria. Saudi J. Humanities Soc Sci, 2025;10(6): 286-293. https://saudijournals.com/media/articles/SJHSS_106_286-293c.pdf

Comment 7: Reviewer 1

If the primary focus is on the relationship between physical inactivity and depression, the inclusion of secondary outcomes (such as obesity or chronic diseases linked to inactivity) seems unnecessary unless directly tied to the study’s main objectives.

Response of Authors:

Thank you for this valuable comment. We acknowledge the importance of maintaining alignment between study objectives and reported outcomes. While the primary aim of our study is to explore the association between physical inactivity and depression among adolescents, we included obesity and chronic diseases in our analysis because they may act as potential confounders in the relationship between physical inactivity and depression. Both obesity and chronic conditions such as diabetes and hypertension have been associated with physical inactivity on one hand, and with poor mental health outcomes, including depression, on the other [1-3]. Therefore, controlling for these factors was necessary to isolate the independent effect of physical inactivity on depressive symptoms.

The following statement is added in the discussion to provide a more comprehensive explanation for the inclusion of chronic diseases and obesity in the analysis of the relationship between inactivity and outcome [lines:321-339]

“Our findings suggest a strong link between physical inactivity, obesity, and depression among adolescents. While the primary objective of this study was to examine the relationship between physical inactivity and depression, the inclusion of obesity and chronic conditions such as diabetes and hypertension was essential, as these variables may act as potential confounders or mediators in this association. Obesity and chronic diseases have been associated with both lower physical activity levels and increased risk of depressive symptoms [32–34]. By accounting for these factors, our analysis provides a deeper understanding of the complex interplay between physical and mental health.

This study indicates that physically inactive adolescents had a 92% higher likelihood of experiencing depression, after adjusting for obesity and chronic diseases. This underscores the independent protective role of physical activity in promoting mental well-being, regardless of existing physical health conditions. Given that adolescents with obesity or chronic diseases are already at elevated risk for depressive symptoms, as was seen in our study and other researchers’ findings (32-35), engaging in regular physical activity may serve as a critical intervention to buffer against these mental health challenges. Physical activity contributes not only to improved physical health but also to enhanced mood, reduced stress, and better emotional regulation, mechanisms that are particularly relevant for youth managing chronic conditions. Therefore, integrating physical activity into daily routines may help adolescents with obesity or chronic illnesses better manage their overall health and reduce the likelihood of developing or worsening depressive symptoms”.

The reference below is incorporated into the manuscript, with appropriate adjustments made to the in-text citation numbering.

32. Castillo F, Francis L, Wylie-Rosett J, Isasi CR. Depressive symptoms are associated with excess weight and unhealthier lifestyle behaviors in urban adolescents. Child Obes. 2014 Oct;10(5):400-7.

33. Kriska A, Delahanty L, Edelstein S, Amodei N, Chadwick J, Copeland K, Galvin B, El ghormli L, Haymond M, Kelsey M, Lassiter C, Mayer-Davis E, Milaszewski K, Syme A. Sedentary behavior and physical activity in youth with recent onset of type 2 diabetes. Pediatrics. 2013 Mar;131(3):e850-6.

34. Alqahtani YA, Shati AA, Alhawyan FS, Alhanshani AA, Al-Garni AM, Al-Qahtani SM, Alshehri MA. Assessment of depression in children and adolescents with Type 1 diabetes mellitus: Impact and intervention strategies. Medicine (Baltimore). 2024 Jul 19;103(29):e38868.

35. Key, J. D., Brown, R. T., Marsh, L. D., Spratt, E. G., & Recknor, J. C. Depressive Symptoms in Adolescents With a Chronic Illness. Children’s Health Care. 2001;30(4), 283–292. https://doi.org/10.1207/S15326888CHC3004_03

Comment one : Reviewer 2

the topic is extremely pertinent and speaks about a worldwide public health crisis

(physical inactivity and adolescent mental well-being) with a special emphasis on the UAE, with sparse data present.

The manuscript points to a startling prevalence of depression and physical inactivity among adol

---

## [Decision Letter · Decision Letter 1]

15 Sep 2025

The Mental Health Impact of Physical Inactivity: A Study on UAE Adolescents.

PONE-D-25-17722R1

Dear Dr. Elgarawany,

We’re pleased to inform you that your manuscript has been judged scientifically suitable for publication and will be formally accepted for publication once it meets all outstanding technical requirements.

Kind regards,

Zulkarnain Jaafar

Academic Editor

PLOS ONE

Additional Editor Comments (optional): 

Reviewer #1: The authors have responded to all the reviewer’s comments.

Reviewer #2: The authors have responded to all the reviewer’s comments.

Reviewer #3: After carefully reviewing the manuscript, even though this topic is relatively common in the existing literature and does not appear to offer a strong element of novelty. However, I note that novel contribution is not a primary requirement for PLOS ONE, as the journal’s main objective is to ensure that the research adds valuable information to the body of evidence. Therefore, rejecting the manuscript solely on the basis of lack of novelty would be unfair to the authors. In addition, while the third reviewer mentioned that the authors did not adequately address the reviewers’ comments, no specific issues were highlighted. Without clear evidence or examples, such a statement seems speculative and more of a personal opinion rather than an objective assessment.

Reviewers' comments:

Reviewer's Responses to Questions

**Comments to the Author**

Reviewer #1: (No Response)

Reviewer #2: All comments have been addressed

Reviewer #3: (No Response)

2. Is the manuscript technically sound, and do the data support the conclusions?

Reviewer #1: (No Response)

Reviewer #2: Yes

Reviewer #3: Partly

3. Has the statistical analysis been performed appropriately and rigorously?

Reviewer #1: (No Response)

Reviewer #2: Yes

Reviewer #3: I Don't Know

4. Have the authors made all data underlying the findings in their manuscript fully available?

Reviewer #1: (No Response)

Reviewer #2: (No Response)

Reviewer #3: No

5. Is the manuscript presented in an intelligible fashion and written in standard English?

Reviewer #1: (No Response)

Reviewer #2: Yes

Reviewer #3: No

Reviewer #1: (No Response)

Reviewer #2: (No Response)

Reviewer #3: After carefully reading it, I find that the topic is rather common in the existing literature and does not offer any novel contribution. I also share many of the concerns raised by the initial reviewers, which I believe the authors have not sufficiently addressed.

**Do you want your identity to be public for this peer review?** For information about this choice, including consent withdrawal, please see our Privacy Policy

Reviewer #1: No

Reviewer #2: No

Reviewer #3: No

---

## [Editor Report · Acceptance letter]

PONE-D-25-17722R1

PLOS ONE

Dear Dr. Elgarawany,

I'm pleased to inform you that your manuscript has been deemed suitable for publication in PLOS ONE. Congratulations! Your manuscript is now being handed over to our production team.

Kind regards,

on behalf of

Dr. Zulkarnain Jaafar

Academic Editor

PLOS ONE